# Flight Trajectory Prediction Based on Automatic Dependent Surveillance-Broadcast Data Fusion with Interacting Multiple Model and Informer Framework

**DOI:** 10.3390/s25082531

**Published:** 2025-04-17

**Authors:** Fan Li, Xuezhi Xu, Rihan Wang, Mingyuan Ma, Zijing Dong

**Affiliations:** 1CAAC Key Laboratory of Flight Techniques and Flight Safety, Civil Aviation Flight University of China, Guanghan 618307, China; xuezhixu@cafuc.edu.cn (X.X.); rihanwang@cafuc.edu.cn (R.W.); 2College of Air Traffic Management, Civil Aviation Flight University of China, Guanghan 618307, China; mamingyuan@cafuc.edu.cn; 3School of Mathematics, Sichuan University, Chengdu 610065, China; zjdong@cafuc.edu.cn

**Keywords:** flight trajectory, ADS-B data, hybrid prediction, interacting multiple model, Informer

## Abstract

Aircraft trajectory prediction is challenging because of the flight process with uncertain kinematic motion and varying dynamics, which is characterized by intricate temporal dependencies of the flight surveillance data. To address these challenges, this study proposes a novel hybrid prediction framework, the IMM-Informer, which integrates an interacting multiple model (IMM) approach with the deep learning-based Informer model. The IMM processes flight tracking with multiple typical motion models to produce the initial state predictions. Within the Informer framework, the encoder captures the temporal features with the ProbSparse self-attention mechanism, and the decoder generates trajectory deviation predictions. A final fusion combines the IMM estimators with Informer correction outputs and leverages their respective strengths to achieve accurate and robust predictions. The experiments are conducted using real flight surveillance data received from automatic dependent surveillance-broadcast (ADS-B) sensors to validate the effectiveness of the proposed method. The results demonstrate that the IMM-Informer framework has notable prediction error reductions and significantly outperforms the prediction accuracies of the standalone sequence prediction network models.

## 1. Introduction

With the increase in air traffic flow and airspace resources stress, effective flight control has become vital for air traffic management (ATM). Based on flight trajectory prediction, more accurate flight control can be implemented, which is conducive to avoiding air conflicts, ensuring flight safety and improving air traffic control (ATC) efficiency. The ATM systems are some of the most important automated air transportation industry systems. ATM system controllers make decisions based mainly on the flight situation in current airspace, the most important of which are flight trajectories, a series of available aircraft current and future position points that provide airspace condition references and accurate, efficient air traffic flow instructions, which supports decision making for ATC. Flight trajectory surveillance is implemented based on real-time aircraft position received from multiple sensors within some geographic areas [1]. Compared with monitoring equipment such as secondary radar, the ADS-B devices based on the global navigation satellite system (GNSS) have significantly improved flight surveillance capabilities with more accurate position information of aircraft [2]. The airborne ADS-B equipment broadcasts relevant information, such as the aircraft’s position, altitude, speed, and identification number, to other aircraft or ground ADS-B sensors without the need for manual operations or inquiries, which allows air traffic controllers to monitor the aircraft’s status. To counter jamming attacks and ensure system reliability, multiple ground sensors are usually established, with the data center receiving multiple ADS-B messages for simultaneous processing [3]. The Department of ATM then uses these data to enforce airspace management and control. The ADS-B air-ground data link is shown in Figure 1.

Due to the large amount of flight position information that can be received with ADS-B sensors, flight trajectory prediction is becoming a core trajectory-based operations (TBO) technology that has drawn a lot of attention [4]. As TBO enables efficient ATC based on 4D trajectory prediction, the International Civil Aviation Organization (ICAO) believes that the TBO is going to be crucial to the Next Generation Air Transportation System (NextGen) [5]; many countries are planning to transform their ATM system to include TBO, for which tracking and surveillance systems and 4D trajectory prediction technologies are critical for resolving congestion and reducing flight conflicts [6]. However, due to the complex environmental and meteorological conditions in the airspace, a variety of flight maneuvers are often required in the critical flight phases, resulting in a change or switch in flight dynamics. The open communication of ADS-B means the received messages are without encryption and integrity protection, and the signal obtained from ADS-B sensors may be interfered with by noise. Moreover, if the ADS-B data update frequency is fixed, or there is a transmission delay, it can lead to an insufficient response in ADS-B data to flight state with the changing dynamics. Therefore, the accuracy of flight trajectory prediction with ADS-B data is often limited. If trajectory predictions are inaccurate, this could lead to errors in the predicted trajectory, which can increase inconsistencies between ATC decision-making and air traffic situations. The common flight trajectory prediction methods with ADS-B data contain two categories [7]. One is to design the recursive optimal filtering or approximate algorithms for real-time flight state prediction, which requires constructing a flight motion dynamic model and combining it with ADS-B measurements. The drawback of these approaches lies in that the output of the filter is often a short-term state estimation, and the estimation accuracy depends on the determined models. Due to the complex air route conditions and the uncertainty of specific flight maneuvers, the single motion model usually fails with changing dynamics. Another kind of method is mainly based on machine learning or deep learning frameworks to develop the time series prediction models for flight trajectory prediction with the rich ADS-B data obtained. However, the long sequences’ dependency is difficult to capture [8]. With the transmission characteristics and the lack of integrity protection, ADS-B data may not accurately respond to the changing flight dynamics, which results in limited trajectory prediction accuracy improvement.

To deal with these deficiencies, this paper proposes a hybrid flight trajectory prediction approach, which integrates interacting multiple model state estimation and the generative deep learning model Informer. The IMM estimator first combines kinematic hypotheses from multiple typical base models to obtain a better estimate of targets with changing dynamics, which is used for the initial state estimation. The estimation correction is then conducted with ADS-B data fusion by exploiting a powerful long sequence forecasting Informer model to generate the predicted errors. With the full use of flight trajectory surveillance data and motion switching modeling, the proposed hybrid trajectory prediction combines the dynamics models-based estimation and data-driven learning framework advantages. Actual ADS-B data are used as experimental data to verify the effectiveness of the proposed algorithm. The comparison results show that the proposed algorithm has more accurate flight trajectory prediction accuracy than the traditional time-series forecasting methods in some critical flight phases.

The remainder of this paper is organized as follows: Section 2 reviews related studies, Section 3 details the proposed IMM-Informer trajectory prediction algorithm, Section 4 outlines our experimental procedures, and the experimental results to demonstrate the performance of the proposed algorithm are illustrated, Section 5 discusses some further comparisons. Finally, Section 6 concludes the paper.

## 2. Related Work

The existing flight trajectory prediction methods focus on the following two aspects:

(1) Dynamics modeling-based predictions. These methods mainly utilize physical-based models combined with flight kinematic quantities and aircraft performance parameters to describe or extrapolate aircraft operational trends. For example, Li et al. [9] generated complete 4D trajectory predictions by fusing the flight state information: position, altitude, speed, and heading of the trajectory feature points. Zhang et al. [10] introduced an aircraft intent model with the aircraft dynamics and kinematics parameters based on aircraft performances to predict trajectories. However, because the parameters required for these methods are often difficult to obtain, there is a risk of significant computational errors. On the other hand, state modeling-based predictions, which conduct state estimation or filtering for flight tracking using the state-space models, have also been widely used as trajectory prediction methods. It is well known that Kalman filtering is the globally optimal linear estimator in minimum mean squared error (MMSE) sense [11], but the accurate state-space model for system dynamics and observation is assumed to be known. For the hybrid system, the state estimation of potentially maneuvering targets often requires multiple filter models to represent varying target behavior. This requirement is achieved with the interacting multiple model estimator [12]. Subsequently, Li et al. [13] designed an interacting multiple model estimator for air target tracking during the flight maneuvers, and Yepes et al. [14] predicted trajectories using a residual-mean interactive multi-model algorithm combined with an intentional model. Although these approaches have achieved success in state estimation for hybrid systems, the improvement of the estimation performance requires accurate dynamics modeling and more prior information, which is often difficult to obtain in advance. Meanwhile, in trajectory prediction in other fields of study, such as human trajectory sensing, autonomous driving, and mobile robots, hybrid filter models can often further improve the state estimation performance to achieve reliable autonomy [15,16]. For example, Nguyen et al. [17] proposed a multimodal fusion LiDAR-inertial odometry method that incorporated the Interactive Multiple Models and Kalman Filter (IMMKF), which can demonstrate superior accuracy for reliable navigation in dynamic motion and noisy conditions.

(2) Machine learning and deep learning-based prediction methods. Machine learning-based prediction methods learn from historical data to extract flight trajectory features, after which a trained model is used for trajectory prediction. For example, Leege et al. [18] used historical trajectory and meteorological data to train trajectory prediction models, which were then used to predict fixed route trajectories and arrival times, and Pang et al. [19] established a stochastic neural network model with Bayesian deep learning for trajectory predictions. To resolve problems associated with the extraction of spatial-temporal features from trajectory data, Ma et al. [20] proposed a hybrid 4D trajectory prediction architecture based on a convolutional neural network (CNN) and long-short-term memory (LSTM) network combination, and Jia [21] proposed attention-LSTM architecture to improve trajectory prediction accuracy, which involved efficiently mining terminal area flight trajectory features; Zhong et al. [22] included the spatio-temporal clustering for historical trajectory data of unmanned aerial vehicles, and proposed a 4D trajectory prediction algorithm to improve the short-term trajectory prediction accuracy. To resolve the unbalanced overall prediction result problems associated with the underutilization of features in machine learning 4D trajectory predictions, Zhao et al. [23] proposed a fractal dimensional feature prediction (FDFP) model based on airborne quick access recorder (QAR) data. To effectively mine the characteristics of the temporal data, Han et al. [24] concluded that the GRU network has significant prediction accuracy advantages. Wang et al. [25] established a flight trajectory feature extraction model based on data attribute association and improved k-means. For the sequence-to-sequence prediction, the Transformer model [26], which uses the multi-head self-attention mechanism to capture the sequences’ dependency, has also been found to better predict aircraft trajectories [27]. Moreover, Zhang et al. [28] proposed a wavelet transform-based prediction framework using multi-scale time-frequency analysis to represent flight trajectory patterns. For multi-sensor fusion in complex environments [29,30], researchers have explored hybrid methods that can better combine the advantages of model-driven and learning-based approaches. Wang et al. [31] proposed an uncertainty-oriented physics-informed LSTM (UOPI-LSTM) network for aircraft dynamic force identification, which demonstrates accurate predictions and strong generalizability even with limited samples.

All in all, 4D trajectory predictions based on deep learning have become a key research focus in recent years, and modern data-driven methods usually model trajectory prediction problems as time series forecasting tasks. However, the model training requires data, and the data quality is subject to actual sensors conditions, which has made the prediction difficult to achieve with high accuracy under complex environments with imprecise models. Therefore, this paper proposes a hybrid 4D trajectory prediction method based on the IMM and Informer model, which combines the dynamics modeling-based state estimation and sequence prediction deep learning framework advantages. The IMM estimation is utilized to predict the 3D trajectory longitude, latitude, and altitude information with changing flight dynamics, after which the Informer model trained with ADS-B data predicts the temporal position deviations to improve prediction accuracy.

## 3. Methodology

### 3.1. Problem Formulation

Flight surveillance data from ground sensors include the entire aircraft flight status information, where the track points are presented as discrete data points, each of which reveals the aircraft position, speed, heading, and other additional flight parameters at specific times [32]. The flight trajectory information can be denoted by Equation (Equation 1),(1)$$t\;p=\left[\begin{array}{c} Lon,Lat,Alt,V,Tra \end{array}\right]$$
where t p denotes the aircraft track point; Lon,Lat,Alt are, respectively, the kinematic quantities of longitude, latitude and aircraft altitude in three dimensions. The actual altitude values used in this study are query normal height, the indicated aircraft airspeed, *V*, and the heading angle, Tra.

Existing deep learning-based flight trajectory predictions are usually formulated as sequence-to-sequence problems. Historical flight trajectory sequences are available as $TP=[tp_{1},tp_{2},...,tp_{t}]$. The trajectory prediction model I(.) can map a current flight trajectory state time length sequence *t* to a 3D position time length sequence *L*, with the predicted model output sequence being the flight trajectory state from time instant t+1 to instant t+L, which is denoted by $FP=\{fp_{t+1},fp_{t+2},...,fp_{t+L}\}$ where $f\;p=\left[\begin{array}{c} Lon,Lat,Alt,t \end{array}\right]$ represents the predicted tracking points. The flight trajectory predictions can be formulated as Equation (Equation 2).(2)$$\left[\begin{array}{c} tp_{1,}tp_{2},...,tp_{t} \end{array}\right]\overset{I(•)}{⟶}\left[\begin{array}{c} fp_{t+1,}fp_{t+2},...,fp_{t+L} \end{array}\right]$$

### 3.2. IMM Estimators

During a transportation flight, control and operation are required to achieve maneuvering in some critical flight phases. Hence, there are some typical kinematic models to describe common aircraft motion processes. However, with the atmospheric environment and complex weather conditions, the flight control process is often accompanied by changing dynamics, which often need to switch between different motion modes. Our proposed method, therefore, makes full use of a priori hybrid systems to model the aircraft’s state motions; the IMM estimators can be exploited as the initial state estimations. Consider the following discrete-time Markov jump stochastic system: (3)$$x_{k}=A\left(r_{k}\right)x_{k-1}+B\left(r_{k}\right)w_{k}$$(4)$$y_{k}=C\left(r_{k}\right)x_{k}+D\left(r_{k}\right)v_{k}$$
where $x_{k}\in R^{n_{x}}$ is the based state; $r_{k}$ is the mode state of finite state Markov chain taking values in $\left\{1,2,...,N_{r}\right\}$; $y_{k}\in R^{n_{y}}$ is the measurement; $w_{k}\in R^{n_{n}}$ is the process noise distributed with $w_{k}∼N\left(w_{k};0,Q_{w}\right)$; $v_{k}\in R^{n}$ is the measurement noise distributed with $v_{k}∼N\left(v_{k};0,R_{v}\right)$. The properly sized matrices A(.),B(.),C(.) and D(.) are known system matrices correspond to the mode state.

Essentially, the IMM estimation is a method for combining state hypotheses from multiple conditioned models to obtain a better state estimate of targets with changing dynamics. Because the single state estimator will exhibit biases when the model is not matched to the target motion, state estimation of potentially maneuvering targets from sensor measurements often requires the use of multiple filter models to account for varying target behavior. The IMM estimators provide effective model management by the interaction and combination among the mode-conditioned state estimates with an underlying Markov chain that controls the switching behavior. Suppose there are previous mode-conditioned estimates ${\left\{x_{k-1|k-1}^{j},P_{k-1|k-1}^{j},\mu_{k-1}^{j}\right\}}_{j=1}^{N_{r}}$. Then, the single IMM algorithmic cycle to obtain current estimator and covariance is as follows.

First, we find the interaction of the mode-conditioned estimates at the beginning of the estimation step with the mixing probabilities ${\left\{\mu_{k-1|k-1}^{ji}\right\}}_{i,j=1}^{N_{r}}$, which are calculated by the following: (5)$${\mu^{ji}}_{k-1|k-1}=\frac{\pi_{ji}{\mu^{j}}_{k-1}}{\sum_{\ell =1}^{N_{r}}\pi_{\ell i}{\mu^{\ell }}_{k-1}}$$
where $\pi_{ji}$ is the transfer probability from model *j* and model *i*. Then, the mixed estimates ${\left\{{\hat{x}}_{k-1|k-1}^{0i}\right\}}_{i=1}^{N_{r}}$ and covariances ${\left\{P_{k-1|k-1}^{0i}\right\}}_{i=1}^{N_{r}}$ can be yielded as(6)$${\hat{x}}_{k-1|k-1}^{0i}=\sum_{j=1}^{N_{r}}\mu_{k-1|k-1}^{ji}{\hat{x}}_{k-1|k-1}^{j}$$(7)$$P_{k-1|k-1}^{0i}=\sum_{j=1}^{N_{r}}\mu_{k-1|k-1}^{ji}\left[P_{k-1|k-1}^{j}+\left({\hat{x}}_{k-1|k-1}^{j}-{\hat{x}}_{k-1|k-1}^{0i}\right){\left({\hat{x}}_{k-1|k-1}^{j}-{\hat{x}}_{k-1|k-1}^{0i}\right)}^{T}\right]$$

Second, mode-conditioned prediction and updating are conducted with the mode-matched filter, that is, for the *i*th model, $i=1,...,N_{r}$, the predicted estimate ${\hat{x}}_{k|k-1}^{i}$ and covariance $P_{k|k-1}^{i}$ are calculated from the mixed estimate ${\hat{x}}_{k-1|k-1}^{0i}$ and covariance $P_{k-1|k-1}^{0i}$ by(8)$${\hat{x}}_{k|k-1}^{i}=A\left(i\right){\hat{x}}_{k-1|k-1}^{0i}$$(9)$$P_{k|k-1}^{i}=A\left(i\right)P_{k-1|k-1}^{0i}A^{T}\left(i\right)+B\left(i\right)Q_{w}B^{T}\left(i\right)$$For the *i*th model, $i=1,...,N_{r}$ calculate the updated estimate ${\hat{x}}_{k|k}^{i}$ and covariance $P_{k|k}^{i}$ from the predicted estimate ${\hat{x}}_{k|k-1}^{i}$ and covariance $P_{k|k-1}^{i}$ as(10)$${\hat{x}}_{k|k}^{i}={\hat{x}}_{k|k-1}^{i}+K_{k}^{i}\left(y_{k}-{\hat{y}}_{k|k-1}^{i}\right)$$(11)$$P_{k|k}^{i}=P_{k|k-1}^{i}-K_{k}^{i}S_{k}^{i}K_{k}^{iT}$$(12)$${\hat{y}}_{k|k-1}^{i}=C\left(i\right){\hat{x}}_{k|k-1}^{i}$$(13)$$S_{k}^{i}=C\left(i\right)P_{k|k-1}^{i}C^{T}\left(i\right)+D\left(i\right)R_{v}D^{T}\left(i\right)$$(14)$$K_{k}^{i}=P_{k|k-1}^{i}C^{T}\left(i\right){\left(S_{k}^{i}\right)}^{-1}$$Then, the updated mode probability $\mu_{k}^{i}$ can be obtained by(15)$$\mu_{k}^{i}=\frac{N\left(y_{k};{\hat{y}}_{k|k-1}^{i},S_{k}^{i}\right)\sum_{j=1}^{N_{i}}\pi_{ji}\mu_{k-1}^{j}}{\sum_{\ell =1}^{N_{r}}N\left(y_{k};{\hat{y}}_{k|k-1}^{\ell },S_{k}^{\ell }\right)\sum_{j=1}^{N_{r}}\pi_{j\ell }\mu_{k-1}^{j}}$$

Finally, the combined estimator ${\hat{x}}_{k|k}$ and covariance $P_{k|k}$ are yielded by using all the mode-matched filters with the mode probabilities: (16)$${\hat{x}}_{k|k}=\sum_{i=1}^{N_{r}}\mu_{k}^{i}{\hat{x}}_{k|k}^{i}$$(17)$$P_{k|k}=\sum_{i=1}^{N_{r}}\mu_{k}^{i}\left[P_{k|k}^{i}+\left({\hat{x}}_{k|k}^{i}-{\hat{x}}_{k|k}\right){\left({\hat{x}}_{k|k}^{i}-{\hat{x}}_{k|k}\right)}^{T}\right]$$

The IMM algorithm in the target tracking task provides more accurate short-term state estimation for the dynamic system with multiple motions. However, the mode-matched optimal filtering needs to use the known state-space model; thus, it is a challenge for the IMM estimation to capture the sequential states with an uncertain dynamical model and cover time-varying features associated with the complex motion patterns. Actually, for the abundance of historical flight surveillance data, the actual prediction errors are available, which can be exploited to generate the future correction errors prediction with a sequence-to-sequence deep learning framework. Therefore, we propose a hybrid method for flight trajectory prediction that introduces an initial IMM output state estimation and combines it with the data-driven prediction correction.

### 3.3. Informer-Based Sequence Prediction

The trajectory prediction task is focused on long sequence time series forecasting (LSTF), which requires greater error prediction capability. To resolve the LSTF problem that hinders it from extending its prediction capability, an efficient Transformer-based model was designed and named Informer [33].

In addition to local time stamps information, hierarchical time stamp information, such as specific hours, minutes, and seconds, is sometimes required [34]. The Informer model uses adaptive input embedding to capture the time series data’s temporal features and has a three-layer feature input representation, as shown in Figure 2.

The three positional embedding representations are categorized into local timestamps, global timestamps, and aligned dimensions, and sine–cosine processing is used to ensure embedding at each track point by fixing the position, as shown in Equations (18) and (19).(18)$$\mathrm{PE}\left(pos,2j\right)=\sin\left(pos/{\left(2L_{x}\right)}^{2j/d_{\mathrm{model}}}\right)$$(19)$$\mathrm{PE}\left(pos,2j+1\right)=\cos\left(pos/{\left(2L_{x}\right)}^{2j/d_{\mathrm{model}}}\right)$$
where $d_{\mathrm{model}}$ denotes the feature dimension. $L_{x}$ denotes the sequence input length. pos denotes the position of the current input trajectory point in the input trajectory sequence and $j\in \{1,...,d_{\mathrm{model}/2}\}$. The global timestamp is represented using a learnable stamp embedding $SE_{\left(pos\right)}$ and one-dimensional convolutional filters are used to map the scalar $x_{i}^{t}$ to a d-dimensional vector $u_{i}^{t}$. The input representation vector for the combination of the positional embedding is shown in Equation (20).(20)$$X_{feed[i]}^{t}=\alpha u_{i}^{t}+PE_{\left(L_{x}\times \left(t-1\right)+i\right)}+\sum_{p}{\left[SE_{\left(L_{x}\times \left(t-1\right)+i\right)}\right]}_{p}$$
where the magnitude between the scalar projection and the local/global embedding if the input is normalized is α=1, and denotes a one-dimensional convolution process to align the dimensions to the vector dimensions, and $i\in \left\{\begin{array}{c} 1,...,L_{x} \end{array}\right\}$.

The ProbSparse self-attention mechanism can be used to effectively deal with long sequence features. Therefore, to improve the overall model performance for the time series prediction tasks, the ProbSparse self-attention mechanism is introduced into the Informer model to improve the long time series self-attention computational efficiency. The ProbSparse self-attention mechanism is a variant of the self-attention mechanism used in the Transformer model. A probabilistic sparse mechanism is introduced to optimize the attention mechanism and capture the importance of different positions in the attention sequence [35,36]. This approach is particularly useful when dealing with long sequences or resource-constrained environments. The ProbSparse self-attention mechanism formulation is shown in Equation (21).(21)$$Attention\left(Q,K,V\right)=\mathrm{Softmax}\left[\frac{\bar{Q}K^{T}}{\sqrt{d}}\right]V$$
where Q,K,V are three matrices of the same size obtained from the linear transformation of the input feature variables, $\bar{Q}$ is obtained from the probabilistic sparsification of *Q*, and Softmax is the activation function. The ProbSparse self-attention mechanism is employed to solve the self-attention problem, with the time complexity computation being optimized from $O(L^{2})$ to $O(L^{*}\log L)$. The multi-head self-attention mechanism is in the same layer as Equation (22) and Equation (23).(22)$$Multihead\left(\begin{array}{c} Q,K,V \end{array}\right)=Concat\left(\begin{array}{c} head_{1},...,head_{n} \end{array}\right)$$(23)$$head_{i}=Attention\left(XW_{i}^{Q},XW_{i}^{K},XW_{i}^{V}\right)$$
where Concat(.) denotes a connection operation, $W^{Q}$, $W^{K}\in R^{d_{\mathrm{model}}\times d_{k}}$, $W^{V}\in R^{d_{\mathrm{model}}\times d_{v}}$, and $d_{k}=d_{v}=d_{\mathrm{model}}/n$. The multi-head attention sends these in parallel to the attention aggregates by making *n* Q, K, and V using different learnable linear projections. To produce the final output, the *n* attention aggregate outputs are then spliced together and transformed using another learnable linear projection. The multi-head ProbSparse self-attention mechanism can generate different queries with different sparsities for each header, which avoids information losses in fixed sparsity patterns [33,37].

The distillation operation shortens each layer’s input sequence lengths, which reduces the number of dimensions and network parameters. Therefore, this framework effectively handles very long input sequences and reduces the memory usage of the stacked layers. After the probabilistic sparsification, the distillation operation enhances the main high-level features and, at the subsequent layer, generates a centralized self-attention feature map, thus compressing the feature dimensions and highlighting the main features [35]. The distillation process from layer *j* to layer *j* + 1 is shown in Equation (24): (24)$$X_{j+1}^{t}=MaxPool\left(ELU\left(Conv1d\left({\left[X_{j}^{t}\right]}_{AB}\right)\right)\right)$$
where ${\left[\cdot \right]}_{AB}$ is the attention block that contains the basic operations in the attention block operations. The distillation method downsamples the features by adding convolutional pooling operations between the neighboring attention blocks. Conv1d(.) denotes a one-dimensional convolutional filter that has a kernel width of 3 and an added ELU activation layer. When x≥0, the function is able to mitigate gradient vanishing, and when x<0, the function is more robust to input variations or noise. The activation function is shown in Equation (25). MaxPool denotes the maximum pooling downsampling operation with a pooling stride of 2. By compressing the features and reducing the number of parameters, the overall memory usage can be reduced to O((2−ε)LlogL).(25)$$ELU\left(x\right)=\left\{\begin{array}{cc} x,\; & if\;x\geq 0\\ \alpha \left(e^{x}-1\right),\; & if\;x<0 \end{array}\right.$$

### 3.4. Hybrid Flight Trajectory Prediction

The proposed hybrid trajectory prediction method combines the IMM and Informer model advantages. To perform the trajectory predictions with changing dynamics, we implement the initial estimates utilizing the IMM algorithm that fuses multiple classical motions. When dealing with sequential flight track data with an unknown model, it is common to use sequence-to-sequence architecture to augment the time series links for the predictions. The informer encoder stage synchronously embeds the spatial state and temporal features to capture the deeper target change patterns in the time series. The ProbSparse self-attention mechanism is crucial for extracting the sequential implicit relationships as it assists the decoder in preserving the temporal dependencies by generating predictions, which further enhances the ability of the long-time-dependent features to characterize the trajectories. IMM is exploited to the multi-model state estimation, which is suitable for scenarios that require switching between different motion modes, and the Informer model is a sequence prediction model based on the ProbSparse self-attention mechanism and self-attention distilling that can deal with long-term time-series dependencies effectively. Therefore, by fusing the informer error outputs with the initial IMM’s multi-model estimation, highly temporal and fine-grained trajectory predictions can be realized. The IMM and Informer combination is used throughout the trajectory prediction process as the two work together for the state estimation and the prediction correction based on time series feature extraction, as shown in Figure 3.

The IMM-Informer hybrid model integrates the IMM framework with the Informer deep learning architecture to provide 4D flight trajectory predictions with ADS-B data. The hybrid method mainly consists of the following steps. Step 1: the IMM framework first mixes the mode-conditioned aircraft’s motion state estimations, each of which is implemented as an optimal filter to estimate the mode-matched state vectors and covariance matrices by utilizing the typical kinematic models. Step 2: These estimations are combined based on updated model probabilities to obtain an initial fused state estimate. According to the historical ADS-B data and IMM estimates, the historical estimation error sequences are available. Step 3: With these historical sequence data, the trained Informer model can generate an error prediction sequence, which utilizes the ProbSparse self-attention and distilling mechanism to capture the long-term temporal dependencies between historical flight surveillance and prediction error sequences. Step 4: The IMM output and the received actual flight surveillance data at the current instant are fed into the Informer model. The Informer’s encoder captures the long-range dependencies, and its decoder generates multi-step future error predictions. Step 5: These refined predictions are reintegrated into the IMM current estimators, where they provide updated state predictions by combining the Informer’s long-term predictions with the IMM’s short-term model-based estimates. The final output is a multi-step prediction of the aircraft’s 4D trajectory, including the latitude, longitude, altitude, and times that correspond to the trajectory. As this hybrid algorithm combines data-driven deep learning with physics-based dynamic models, it effectively addresses the trajectory prediction challenges of fusing flight surveillance data with high uncertainty and varying motion dynamics. The complete cycle of the IMM-Informer algorithm is summarized in Algorithm 1.
**Algorithm 1:** Hybrid Flight Trajectory Prediction Algorithm**Input the IMM previous estimators**: obtain the previous mode-conditioned state estimation ${\hat{x}}_{k-1|k-1}^{j}$, covariance matrices $P_{k-1|k-1}^{j}$ and mode probabilities $\mu_{k-1}^{j}$. The priori transition probability is given as $\pi_{ij}$ for switching from mode *i* to mode *j*.**State interaction**:$\mu_{k-1|k-1}^{ji}=\frac{\pi_{ji}\mu_{k-1}^{j}}{\sum_{l=1}^{N_{r}}\pi_{li}\mu_{k-1}^{l}},$${\hat{x}}_{k-1|k-1}^{0i}=\sum_{j=1}^{N_{r}}\mu_{k-1|k-1}^{ji}{\hat{x}}_{k-1|k-1}^{j},$$P_{k-1|k-1}^{0i}=\sum_{j=1}^{N_{r}}\mu_{k-1|k-1}^{ji}\left[P_{k-1|k-1}^{j}+\left({\hat{x}}_{k-1|k-1}^{j}-{\hat{x}}_{k-1|k-1}^{0i}\right){\left({\hat{x}}_{k-1|k-1}^{j}-{\hat{x}}_{k-1|k-1}^{0i}\right)}^{T}\right].$**Mode-matched prediction update**: calculate estimate ${\hat{x}}_{k|k-1}^{i}$ and covariance $P_{k|k-1}^{i}$.**Mode-matched measurement update**: calculate ${\hat{y}}_{k|k-1}^{i},S_{k}^{i},K_{k}^{i},{\hat{x}}_{k|k}^{i},P_{k|k}^{i},\mu_{k}^{i}$.**Combination State estimate weighted by updated probabilities**:${\hat{x}}_{k|k}=\sum_{i=1}^{N_{r}}\mu_{k}^{i}{\hat{x}}_{k|k}^{i},\;P_{k|k}=\sum_{i=1}^{N_{r}}\mu_{k}^{i}\left[P_{k|k}^{i}+\left({\hat{x}}_{k|k}^{i}-{\hat{x}}_{k|k}\right){\left({\hat{x}}_{k|k}^{i}-{\hat{x}}_{k|k}\right)}^{T}\right]$.**Sequence feature extraction and error prediction**: input the historical data and IMM state prediction, output error sequence with Informer:**Encoder**: $\mathrm{Enc}(X_{k|k,l}),$ where $X_{k|k,l}≜\left[y_{k-l},\cdots ,y_{k-1},y_{k},{\hat{x}}_{k|k}\right]$**Decoder** generate error prediction: ${\hat{Y}}_{\mathrm{Informer}}=\mathrm{Dec}\left(\mathrm{Enc}\left(X_{k|k,l}\right)\right)$**Correction fusion**: combine error prediction ${\hat{Y}}_{\mathrm{Informer}}$ and IMM estimates ${\hat{X}}_{k|k}$:${\tilde{X}}_{k|k}={\hat{Y}}_{\mathrm{Informer}}+{\hat{X}}_{k|k},$ where ${\hat{Y}}_{\mathrm{Informer}}≜\left[\hat{\Delta x_{k}},\cdots ,\hat{\Delta x_{k+L}}\right]$, ${\hat{X}}_{k|k}≜\left[{\hat{x}}_{k|k},\cdots ,{\hat{x}}_{k+L|k+L}\right]$**Output**: return prediction ${\tilde{X}}_{k|k}$.

## 4. Experiments

### 4.1. Data Acquisition and Preprocessing

The dataset used in this paper is the actual flight surveillance data received from ADS-B sensors for multiple flights between two large civil airports in southwestern China over one month in 2024. The dataset has relatively complete departure and arrival flight trajectories, which include data from the time the aircraft takes off to the time it lands. Collected by multiple sensors, the received raw ADS-B messages contain real-time aircraft position information (including latitude, longitude, and altitude) and other flight status information. After ADS-B messages decoding and parsing, the data are stored in packets with the fixed time intervals (e.g., every 1 or 2 s); the data are stored as CSV files that contain multi-dimensional information, such as kinematic quantities of position, velocity, and headings, with each parameter type being recorded in a fixed column of the file; therefore, there is a consistent data structure for subsequent processing and analysis. The flight surveillance data used here are the time series data from the ADS-B messages received by the sensor after decoding. The final trajectory data selected as the input includes six dimensions: timestamp, position (latitude, longitude, altitude), ground speed, and heading. The time interval is 1s, and the flight times are about 3 h; thus, there are about $10^{4}$ track points in one whole flight.

When acquiring the decoded ADS-B data, there may be some abnormal data points due to issues such as signal interference and terrain occlusion with the placement of ADS-B sensors in certain geographic areas. Therefore, it is necessary to first eliminate these abnormal data points and perform simple interpolation and return the cleaned dataset. Then, ADS-B data from multiple sensors are synchronized via timestamp matching (±1 s tolerance). As the dynamical system modeling in this paper uses the position, velocity, and acceleration of the aircraft in the Cartesian coordinate system as the track state vectors in the kinematic equations, it is necessary to use the Mercator projection method to convert the latitude and longitude of each trajectory point in the WGS84 coordinate system to a position in the Cartesian coordinate system and form the 3D position coordinates with altitude. The data then need to be normalized to improve the training speed and reduce the influence of different data units and the computational complexity. Normalization requires the data to be mapped onto a [0, 1] interval so that the features are in the same order of magnitude, which allows for an effective comparison between the data and facilitates faster data preprocessing. As a common normalization method, Min-Max is expressed as(26)$$x^{*}=\frac{x-x_{\min}}{x_{\max}-x_{\min}}$$
where $x^{*}$ is the normalized data, *x* is the raw data, $x_{\max}$ is the maximum value of the data and $x_{\min}$ is the minimum value of the data.

### 4.2. Experimental Setting and Environment

To describe the operational state of the aircraft, the typical models for target tracking are exploited to represent the dynamics-based physical state motion process. Three common kinematic models: constant velocity (CV), constant acceleration (CA) and constant turn (CT), which are selected and combined with the aircraft status data, are used in this example. However, the dynamic system model reflecting the real flight trajectory is difficult to determine in advance. In order to deal with the kinematic models’ uncertainty, IMM estimation is firstly used for initial state estimation in the proposed hybrid flight trajectory prediction. The IMM framework is combined with Bayesian recursive estimations between the different models using parallel operations, which enables the fusion of the multi-model state estimates and covariance. This combination means that the algorithm is more adaptable in response to flight dynamic changes. Considering the multiple operations during the flight, the changing dynamics are modeled by the modal state Markov chain. That is, the aircraft motion can be represented through mutual switching between the three motion models.

In this study, the target motion is represented in a 3D Cartesian coordinate system (ξ−η−γ), where ξ, η and γ, respectively, denote the target’s horizontal, lateral, and vertical positions. Different modes with the transition probabilities are set as $\pi_{ii}=0.8$ for i=1,2,3, and $\pi_{ij}=0.1$ for i≠j, and these motion models are defined in the following:(1)CV Model

The CV model assumes that the target moves at a constant velocity without acceleration, which is suitable for steady climb and cruise conditions. The state vector for the CV model is defined as $x={\left[\xi ,\dot{\xi },\eta ,\dot{\eta },\gamma ,\dot{\gamma }\right]}^{T}$, where ξ,η,γ is the position, $\dot{\xi },\dot{\eta },\dot{\gamma }$ are the velocity components in the Cartesian coordinate system. The state transition equation can be described by the following: (27)$$\begin{array}{c} x_{k+1}=\left[\begin{array}{cccccc} 1 & T & 0 & 0 & 0 & 0\\ 0 & 1 & 0 & 0 & 0 & 0\\ 0 & 0 & 1 & T & 0 & 0\\ 0 & 0 & 0 & 1 & 0 & 0\\ 0 & 0 & 0 & 0 & 1 & T\\ 0 & 0 & 0 & 0 & 0 & 1 \end{array}\right]x_{k}+w_{k} \end{array}$$
where T=1 s is the time interval, and the noise process is modeled as white noise with the variance given as follows:(28)$$Q_{w}=\left[\begin{array}{ccc} {\bar{Q}}_{CV} & 0 & 0\\ 0 & {\bar{Q}}_{CV} & 0\\ 0 & 0 & {\bar{Q}}_{CV} \end{array}\right],\;\mathrm{and}\;{\bar{Q}}_{CV}=q\left[\begin{array}{cc} T^{4}/4 & T^{3}/2\\ T^{3}/2 & T^{2} \end{array}\right]$$

(2)CA Model

The state vector for the CA model is defined as $x={\left[\xi ,\dot{\xi },\ddot{\xi },\eta ,\dot{\eta },\ddot{\eta },\gamma ,\dot{\gamma },\ddot{\gamma }\right]}^{T}$, where $\ddot{\xi },\ddot{\eta },\ddot{\gamma }$ are the acceleration components. The state motion equation is given by the following: (29)$$x_{k+1}=\left[\begin{array}{ccc} F & 0 & 0\\ 0 & F & 0\\ 0 & 0 & F \end{array}\right]x_{k}+w_{k}$$
where $F=\left[\begin{array}{ccc} 1 & T & T^{2}/2\\ 0 & 1 & T\\ 0 & 0 & 1 \end{array}\right]$.

(3)CT Model

The motion equation for the CT model is given by the following: (30)$$\begin{array}{c} x_{k+1}=\left[\begin{array}{cccc} 1 & \frac{\sin(\omega T)}{\omega } & 0 & -\frac{1-\cos(\omega T)}{\omega }\\ 0 & \cos(\omega T) & 0 & -\sin(\omega T)\\ 0 & \frac{1-\cos(\omega T)}{\omega } & 1 & \frac{\sin(\omega T)}{\omega }\\ 0 & \sin(\omega T) & 0 & \cos(\omega T) \end{array}\right]x_{k}+w_{k} \end{array}$$
where $x_{k}={\left[\xi_{k},{\dot{\xi }}_{k},\eta_{k},{\dot{\eta }}_{k}\right]}^{T}$, ω is the rate of turn calculated from state. The noise variance matrix for this model has the following form:(31)$$Q_{w}=\left[\begin{array}{cc} {\bar{Q}}_{CT} & 0\\ 0 & {\bar{Q}}_{CT} \end{array}\right],\;\mathrm{and}\;{\bar{Q}}_{CT}=q\left[\begin{array}{cc} T^{3}/3 & T^{2}/2\\ T^{2}/2 & T \end{array}\right]$$

The Informer model was trained and tested on a server with a 64-bit operating system Windows 11, a single NVIDIA RTX4090 24 GB GPU. To ensure compatibility with the required software tools and libraries, the analysis and development programming language was Python 3.7. The optimal model architecture and parameter settings were obtained through continuous trial and error experiments. Therefore, the model’s hyperparameter settings for this experiment, which were determined through a grid search, are shown in Table 1. The encoder layer for the Informer is selected from {6,4,3,2}, with the decoder layer set at 2. The number of heads for the multi-head attention is selected from {8,16}, and the multi-head attention output dimension is set at 512. A 3-layer stack is included in the encoder with a 2-layer decoder. As the decoder start token is a truncated segment from the encoder input sequence, the decoder start sequence length must be less than the encoder input length. Our proposed method is optimized using the Adam optimizer with the learning rate starts from 1 × 10^−4^, decay factor 0.1 and step size 2 of the total epochs 20. The batch size is set at 64. In each epoch, the model performs a complete traversal over the training data with the teacher forcing strategy and calculates the model loss based on the evaluation metrics, after which the optimizer is used to update the model parameters. The Early Stopping is used to select the optimal model parameters with the validation set, where the monitoring indicator patience is set 5, the min_delta is set to 1 × 10^−4^.

### 4.3. Evaluation Metrics

The mean absolute error (MAE), root mean square error (RMSE) and mean absolute percentage error (MAPE) are used to measure the state estimation errors. The MAE, RMSE, MAPE are defined as(32)$$\mathrm{MAE}=\frac{1}{n}\sum_{i=1}^{n}\left|\hat{y}-y\right|$$(33)$$\mathrm{RMSE}=\sqrt{\frac{1}{n}\sum_{i=1}^{n}{\left(\hat{y}-y\right)}^{2}}$$(34)$$\mathrm{MAPE}=\frac{1}{n}\sum_{i=1}^{n}\left|\frac{\hat{y}-y}{y}\right|\times 100\%$$
where *y* and $\hat{y}$ are the actual and predicted positions on the flight track at time instant *i*.

In the time series analyses, these metrics effectively measure the model performance based on the errors between the actual and predicted positions, with smaller error metric values indicating higher model prediction accuracy.

### 4.4. Experimental Results

In this study, the selected flight phases for experimental verification are climb, descent, and approach and landing, though the complete trajectory from aircraft take-off to landing is used. The climb phase extends from the end of the take-off (ascending) segment (about 450 m) to below cruise altitude; the descent phase extends from the descent with initial approach start altitude to about 450 m, which corresponds to the climb phase, and the landing phase contained final approach extends from an altitude less than 450 m, at which time the aircraft drops its flaps, lowers the landing gear, aligns to the runway, and landing starts at around 15.3 m above the ground, when the aircraft safely lands, stops, and reaches zero speed.

To test the prediction validity of the proposed algorithm, the typical data-driven time series prediction deep learning model is used for comparison. LSTM is a special recurrent neural network that addresses gradient vanishing and explosion problems through the introduction of gating mechanisms, enabling it to capture long-term dependencies in sequences. Informer, like the Transformer, employs self-attention mechanisms and parallel computation to capture global dependencies in sequences. Unlike LSTM’s autoregressive generation requiring iterative steps, Informer processes entire sequences as a whole. Informer is more efficient than the traditional Transformer for the LSTF tasks due to its ProbSparse self-attention mechanism and self-attention distilling mechanisms. Since LSTM and Informer are representative, they are chosen for comparison with the proposed method. Both models are characterized by their respective longitudes, latitudes, altitudes, and corresponding times, with the other hyper-parameters set similarly to the IMM-Informer model. All the models were trained on the ADS-B data for the same flight. The randomly selected 24 days of the receiving data are the training set, the remaining 4 days are the validation set, and the last 2 days are the test set. The predicted aircraft trajectories in the three flight climb, descent, approach and landing phases are shown on the time axis in Figure 4a–c for longitude, latitude, and altitude.

As shown in Figure 4, the predicted trajectories with the proposed IMM-Informer are more accurate than the flight trajectories predicted by other models in all phases, are very similar to the actual trajectories, and have fewer fluctuations. While the Informer model and the LSTM model have the same initial trend as the IMM-Informer architecture, they begin to have larger deviations over time. The Informer model has the worst longitude, latitude, and altitude data predictions compared to the other models. The proposed method makes full use of the multiple dynamics models to capture the kinematic switching; using this as the initialization of the data-driven method, combined with historical data to train the trajectory prediction model, the prediction performance has been improved in key phases compared with the deep learning framework alone. In addition, the comparison results for these prediction methods are shown in Table 2 along with the three evaluation metric analyses, MAE, RMSE, and MAPE, for the predicted and actual latitude, longitude, and altitude values with the different flight phases.

Based on Table 2, it can be concluded that the IMM-Informer architecture single longitude, latitude, and altitude predictions outperform the other models, followed by the LSTM model. Compared to the LSTM model, in the climb phase, the IMM-Informer architecture MAE values reduce by 0.08, 0.05, and 45.62, and the RMSE values, respectively, reduce by 0.07, 0.04, and 51.07. Similarly, in the descent phase, the model’s MAE values, respectively, reduce by 0.10, 0.09, and 75.66 for longitude, latitude, and altitude, and the RMSE values reduce by 0.11, 0.09, and 77.29. In the approach and landing phase, the model, respectively, reduces the MAE longitude, latitude, and altitude values by 0.09, 0.06, and 50.97 and the RMSE values by 0.09, 0.07, and 75.04. Compared to the other models, the IMM-Informer architecture minimizes the longitude, latitude, and altitude predictions in all three metrics. For actual ATC system, accurate flight path prediction can present the current and future air operation situation, and provide decision making basis for the identification potential air conflicts and optimization path by controllers.

In this experiment, complete trajectories are used to compare the latitude, longitude, and altitude prediction effects in the different phases with different flights; the results are given in Figure 5a,b. From the overall 3D trajectory prediction comparisons, the IMM-Informer and LSTM models predict trajectories that are the closest to the actual trajectories, followed by the Informer model. Then, based on the comparison of the actual trajectories and the analysis of the three MAE, RMSE, and MAPE evaluation indexes, we compare the integrated error value metrics in Table 3.

Table 3 compares the evaluation metrics for different flights (a) and (b), from which the following conclusions can be drawn. The combined MAE, RMSE and MAPE metric analyses show that the IMM-Informer algorithm trajectory prediction errors are much lower than those for the LSTM and Informer models in all three flight phases. While there are some errors in the IMM-Informer algorithm, these are minimal compared to the other models. Compared to the LSTM, in all three phases, the respective MAE values averagely reduce by 24.21, 36.31 and 28.11, and the RMSE values averagely decline by 52.11, 74.26, and 74.39. The IMM-Informer algorithm’s MAPE values are decreased 0.58, 0.48, and 1.33 by mean, which indicates the feasibility and accuracy of the IMM-Informer architecture in predicting trajectories. Because the proposed IMM-Informer architecture is more effective and accurate, it better meets the trajectory prediction task aircraft requirements by contrast.

## 5. Discussion

To verify the generalization and computational efficiency of the proposed method, we extend the dataset and baseline methods. The training dataset has been expanded by the flight surveillance data of multiple flights with different routes and the flight trajectories generated by the stochastic system model, which account for about 30% and 10%, respectively. The other parameters of the model training were consistent with the previous experiment. As the proposed flight trajectory prediction approach combines dynamics modeling and learning-based methods, we extend three sequence prediction-oriented deep learning networks model and the classical filtering method as added baselines to show the performance of our method. The details of these baselines are summarized as follows.

(1)KF: The traditional Kalman filtering method [11] with a single motion model is used for state estimation.(2)IMM-LSTM: IMM is used for initial state prediction, and LSTM [38] is used for error sequence prediction.(3)IMM-GRU: IMM is used for initial state prediction, and the Gated Recurrent Unit (GRU) [39] network is used for error sequence prediction.(4)IMM-Transformer: IMM is used for initial state prediction and Transformer [26] is used for error sequence prediction.

Table 4 lists the results with the best results highlighted.

Table 4 compares the performance of the proposed method with six other trajectory prediction methods in three key flight phases (climb, descent, approach and landing). The evaluation metrics MAE, RMSE and MAPE show that the overall performance of the hybrid methods are better than that of the single learning-based sequence prediction method and the traditional Kalman filtering. In addition, although the training set contains flight surveillance data of different routes, the proposed IMM-Informer can demonstrate significant advantages in all phases, especially in the approach and landing phases with dynamic switching scenarios. Therefore, the effectiveness and generalization of the proposed hybrid flight trajectory prediction with IMM and Informer are verified in our experiment. Moreover, the detailed error fading across the flight phases and the confidence intervals with the proposed hybrid prediction method are shown in Figure 6, where the predicted trajectory 90% confidence intervals with the proposed method are shown in yellow.

As can be seen from Figure 6, taking the climbing stage as an example, the confidence intervals of the proposed method contains the real trajectory, and the hybrid methods using multi-model information are superior to the single data-driven method and the traditional Kalman filtering, although the Kalman filtering also uses the motion model. The reason lies in that a single model cannot effectively represent the dynamic changes or uncertainties existing in the flight process. In the hybrid prediction methods, IMM fused with LSTM and GRU, which are based on recurrent neural network architecture, have a similar prediction performance, but the overall performances are lower than the hybrid method based on Transformer architecture because the latter utilizes the self-attention mechanism to capture the sequence dependence. In addition, compared with the proposed IMM-Informer, the prediction performance of IMM-Transformer fluctuates at some points. The reason may lie in the global attention mechanism of the Transformer being more sensitive to local dependence. Also, there are many variants of the Transformer structure for specific tasks, and Informer achieves an effective balance between longer sequence predictions and local dependencies. Actually, the proposed hybrid method combined with dynamic modeling and being data-driven can be viewed as knowledge embedding on the flight surveillance data, and the algorithm fusion is conducted based on the sensor surveillance data. In order to represent more complex environments, such as complex weather, routes, sensor anomalies and packets loss, more prior knowledge is needed to build models for interaction, and the models selected and the construction of the general model will be another type of challenge.

To discuss the feasibility of deploying the proposed method in actual ATC systems, the efficiency comparison of the hybrid methods with different flight trajectory prediction phases is presented. We evaluate model efficiency concerning both training and testing times. The efficiency comparison results are given in Figure 7.

It can be seen from Figure 7 that although the Transformer architecture-based learning methods need more time than the recurrent neural network method in the model training process with the same training data and epochs, the proposed flight trajectory prediction based on IMM-Informer takes less than 3 s in the test dataset. For the common flight monitoring equipment, the sampling frequency of 1s is used. Therefore, the proposed hybrid method has the real-time feasibility of deploying the module in the actual ATC system. The development of the specific application system will adopt the way of offline learning and online prediction, which is also the common architecture of the existing intelligent monitoring and early warning system.

## 6. Conclusions

Accurate trajectory prediction not only improves flight safety and reduces accident rates but also optimizes route design and reduces unnecessary flight conflicts. To address data-driven flight trajectory predictions with ADS-B data, this study proposes a hybrid trajectory prediction model based on IMM and Informer architecture. The IMM model combines multiple motion models to flexibly cope with complex dynamic environments by kinematic switching, and the Informer model learns from the historical trajectory data and generates the correction sequence to modify prediction trajectories at multiple future moments. The Informer structural design also allows for better sequences’ dependency alignment when dealing with long sequence time-series forecasting, which significantly improves the efficiency of generating correction sequences. The MAE, RMSE, and MAPE evaluations of the models’ performances show that the IMM-Informer model has excellent trajectory prediction performances and can effectively deal with dynamic changes and uncertainties. Because the model harnesses the advantages of fusion state estimation with multiple motion models and error correction, the biases that may occur using imprecise models or standalone flight surveillance data are avoided.

The advancements in the proposed flight trajectory prediction method enable the early identification of potential conflicts, reducing bypass flights in dense airspace, thereby indirectly enhancing airspace capacity and providing air traffic controllers with critical decision-making support. Furthermore, by integrating precise trajectory predictions with TBO, the framework assists controllers in optimizing flight sequences to minimize engine idle fuel consumption, offering substantial potential for fuel efficiency and emission reduction. However, challenges remain. The abnormal existence of ADS-B data will affect the initial state prediction with the prior model. Complex and special flight dynamics will also extend the base models to accommodate additional sensor inputs. How to design more effective fusion methods with general dynamics modeling or representation in flight trajectory prediction based on the hybrid framework and the effective knowledge embedding of flight surveillance data with abnormalities will be further studied.

## Figures and Tables

**Figure 1 sensors-25-02531-f001:**
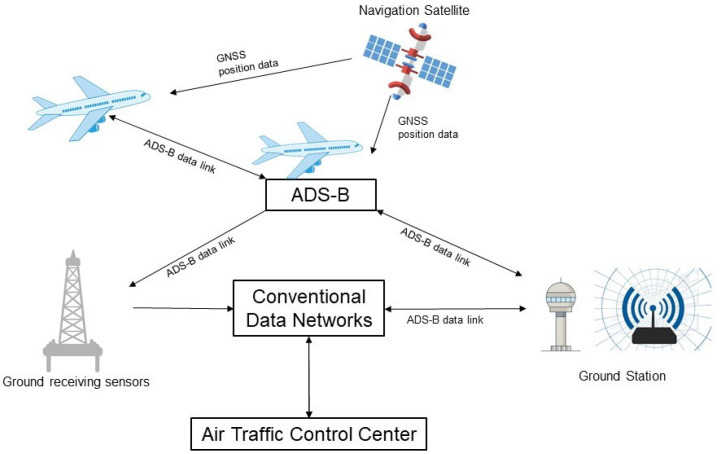
ADS-B data transfer with sensors.

**Figure 2 sensors-25-02531-f002:**
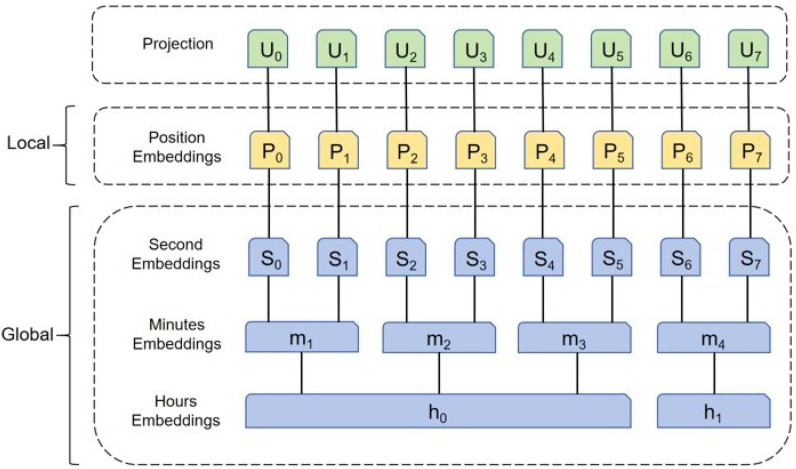
The embedding of input in Informer framework.

**Figure 3 sensors-25-02531-f003:**
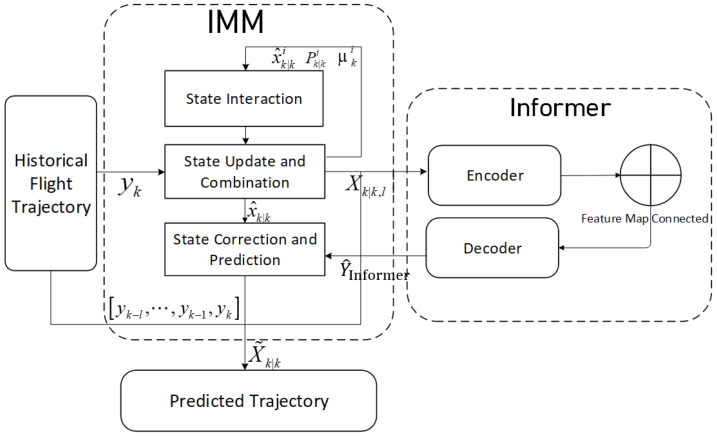
Flight trajectory prediction model based on IMM-Informer.

**Figure 4 sensors-25-02531-f004:**
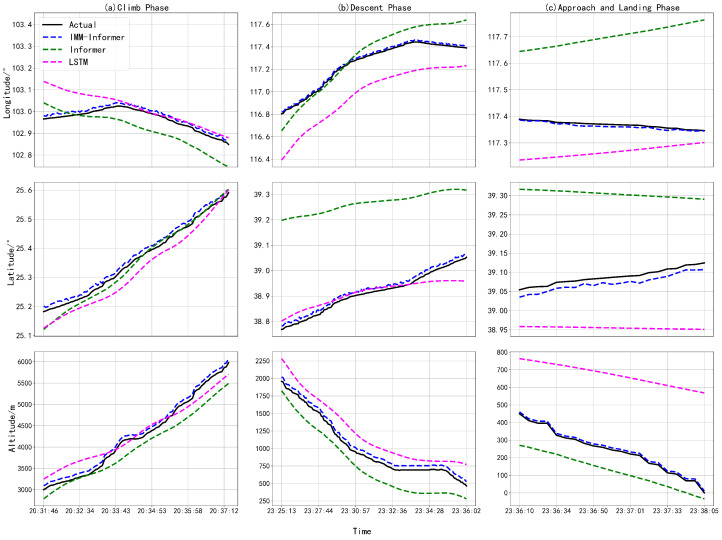
Predicted latitude, longitude and altitude with the time for three phases.

**Figure 5 sensors-25-02531-f005:**
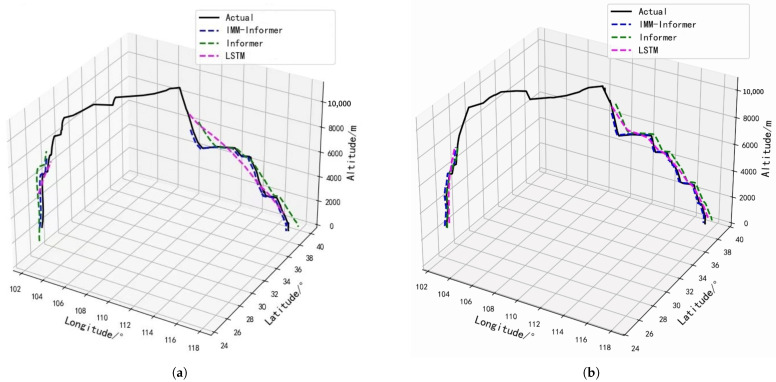
Trajectory predictions for two different flights (**a**,**b**) in three-dimensional coordinates.

**Figure 6 sensors-25-02531-f006:**
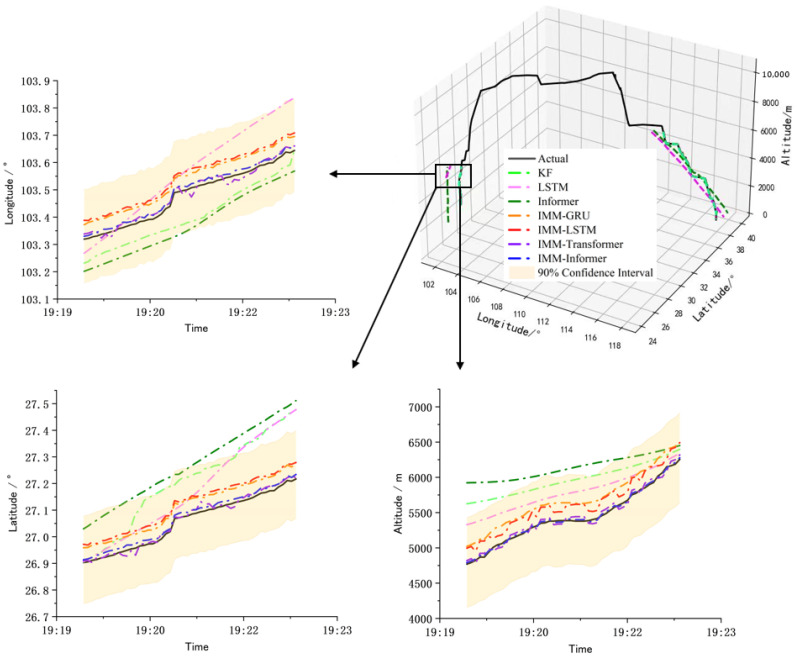
The prediction results comparison of the climb phase with different methods.

**Figure 7 sensors-25-02531-f007:**
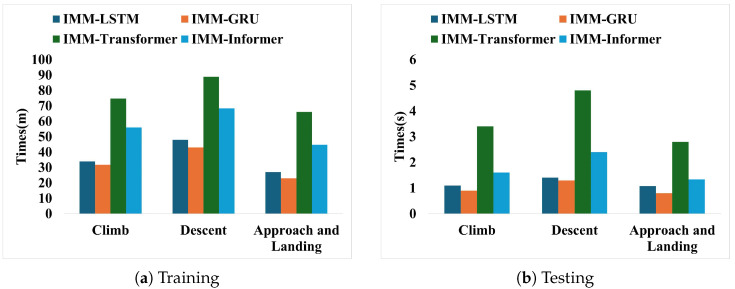
Efficiency comparison for different hybrid methods.

**Table 1 sensors-25-02531-t001:** Model parameter settings.

Hyperparameter	Parameter Value
epoch	20
d_model	512
num_heads	8
activation	GeLU
e_layers	3
d_layers	2
d_ff	512
dropout_rate	0.1
output_size	3
initial learning rate	1 × 10^−4^
seq_length	30
batch size	64
optimizer	Adam

**Table 2 sensors-25-02531-t002:** Evaluation metrics comparison results.

Flight Phase	Evaluation Indicator	Model	Latitude/°	Longitude/°	Altitude/m
Climb Phase	MAE	Informer	0.99548	0.03444	132.06817
		LSTM	0.10343	0.06473	86.02405
		IMM-Informer	0.02852	0.01891	40.40406
	RMSE	Informer	1.00458	0.04582	192.69011
		LSTM	0.12058	0.07505	101.95902
		IMM-Informer	0.04233	0.03108	50.88601
	MAPE	Informer	0.83718	0.12060	4.61993
		LSTM	0.08680	0.22175	2.09658
		IMM-Informer	0.02410	0.06523	0.13332
Descent Phase	MAE	Informer	0.41380	0.12588	136.25251
		LSTM	0.13580	0.10085	110.34593
		IMM-Informer	0.03016	0.01398	34.68435
	RMSE	Informer	0.42711	0.14870	165.37633
		LSTM	0.17413	0.13184	137.15522
		IMM-Informer	0.06864	0.04478	59.86134
	MAPE	Informer	0.38872	0.48639	2.39682
		LSTM	0.12810	0.38845	1.87864
		IMM-Informer	0.02835	0.05392	0.067292
Approach and Landing Phase	MAE	Informer	0.70733	0.07791	124.84474
		LSTM	0.11727	0.07358	90.75797
		IMM-Informer	0.02918	0.01400	39.77841
	RMSE	Informer	0.76103	0.10420	180.26527
		LSTM	0.14611	0.10011	139.50289
		IMM-Informer	0.05531	0.03468	64.45676
	MAPE	Informer	0.61710	0.29185	5.18018
		LSTM	0.10547	0.27147	6.13991
		IMM-Informer	0.02612	0.050081	0.07094

**Table 3 sensors-25-02531-t003:** Prediction performance comparison with different flights.

Flight Phase	Model	MAE/m	RMSE/m	MAPE/%
Climb Phase (a)	Informer	45.47609	107.35282	1.72353
	LSTM	26.63042	56.84638	1.21374
	IMM-Informer	13.46547	16.24835	0.52853
Descent Phase (a)	Informer	47.46290	97.74983	1.18429
	LSTM	38.28472	81.37840	1.00298
	IMM-Informer	14.37539	16.28472	0.42749
Approach and Landing (a)	Informer	37.45782	106.12834	4.92387
	LSTM	32.27865	82.34620	2.26375
	IMM-Informer	13.10498	17.24801	0.83038
Climb Phase (b)	Informer	44.36603	111.25126	1.85924
	LSTM	28.73074	58.86615	0.90171
	IMM-Informer	13.48383	15.24737	0.44741
Descent Phase (b)	Informer	45.59741	95.48042	1.09064
	LSTM	36.86086	79.18678	0.89841
	IMM-Informer	11.15761	15.76549	0.52517
Approach and Landing Phase (b)	Informer	41.87668	104.07714	5.06006
	LSTM	30.31627	80.54209	2.04663
	IMM-Informer	14.27386	16.87389	0.82993

**Table 4 sensors-25-02531-t004:** Prediction performance comparison with extended dataset.

Method	Climb Phase			Descent Phase			Approach & Landing Phase		
	**MAE/m**	**RMSE/m**	**MAPE/%**	**MAE/m**	**RMSE/m**	**MAPE/%**	**MAE/m**	**RMSE/m**	**MAPE/%**
KF	45.43819	104.38423	1.63842	42.39424	90.34294	1.12782	42.38492	102.384209	3.58538
Informer	46.45449	105.46542	1.76324	46.46313	95.56364	1.25564	37.45782	104.43744	4.75356
LSTM	37.67862	54.78643	1.23545	37.45215	82.45632	1.04563	38.27865	81.457477	2.23447
IMM-LSTM	17.58695	26.46275	0.74762	16.56773	26.64336	0.78546	17.33497	26.75425	1.36434
IMM-GRU	17.86972	26.67356	0.76854	16.86354	26.72543	0.77644	17.23485	26.76256	1.27642
IMM-Transformer	9.67325	12.56634	0.59768	10.55768	13.65266	0.56746	12.81507	14.35884	0.94563
IMM-Informer	**7.46525**	**9.24632**	**0.51648**	**8.37539**	**9.56732**	**0.45567**	**10.10534**	**12.17654**	**0.82853**

## Data Availability

All data and codes that support the findings of this study are available from the corresponding author upon reasonable request.

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
