# Peer review of "Flight Trajectory Prediction Based on Automatic Dependent Surveillance-Broadcast Data Fusion with Interacting Multiple Model and Informer Framework"

_sensors, 2025, doi:10.3390/s25082531_

Round 1
Reviewer 1 Report
Comments and Suggestions for Authors
This paper presents a hybrid flight trajectory prediction model that integrates the Interacting Multiple Model (IMM) estimator with the deep learning-based Informer framework. The proposed IMM-Informer approach aims to improve the accuracy and robustness of aircraft trajectory predictions using Automatic Dependent Surveillance-Broadcast (ADS-B) data. The methodology is well-structured, and the hybrid model leverages the strengths of both physics-based dynamic modeling and data-driven learning. The experimental results demonstrate significant error reductions compared to conventional time-series forecasting models. While the study is valuable, some aspects require further refinement as outlined below.
Major Comments:
1.Clarity of Methodology:
The explanation of the IMM estimation process in Section 3.2 is relatively technical and may be difficult for readers unfamiliar with multiple model estimation techniques. It is recommended to provide a brief intuitive explanation before diving into the mathematical formulation.
The fusion strategy between IMM and Informer in Section 3.4 could be better structured. While the method is explained in detail, a flowchart illustrating the step-by-step prediction process would significantly enhance clarity.
2.Comparative Analysis:
The study compares IMM-Informer with LSTM and standalone Informer models. However, it would be beneficial to include another baseline, such as a traditional Kalman filter-based approach, to further highlight the advantage of the hybrid methodology.
The justification for selecting LSTM and Informer as the primary deep learning baselines is not clearly stated. A brief discussion on why other deep learning models (e.g., Transformer-based models beyond Informer) were not considered would strengthen the experimental section.
3.Evaluation Metrics and Result Interpretation:
While the experimental results in Table 2 and Table 3 show that the proposed approach outperforms the baselines, the magnitude of improvement in some phases is relatively small. A discussion on the practical significance of these improvements for air traffic management applications is suggested.
The MAE, RMSE, and MAPE metrics are appropriate for evaluating trajectory prediction accuracy, but additional statistical significance testing (e.g., paired t-tests or confidence intervals) could further confirm the robustness of the performance gains.
Minor Comments:
1.Abstract Length:
The abstract is informative but slightly lengthy. Consider streamlining the explanation of the methodology while retaining key results to improve conciseness.
2.Equation Formatting Issues:
Please check whether there are formatting inconsistencies around Equation (21) in Section 3.3. The indentation or spacing might need adjustments for better readability.
3.Figures and Tables:
The trajectory prediction results in Figure 4 effectively visualize the model's performance. However, adding a zoomed-in comparison of key regions where the predictions diverge from the actual trajectory could provide more insight.
4.Conclusion Section:
The conclusion effectively summarizes the study but could briefly mention potential real-world deployment challenges and future research directions, such as handling ADS-B data anomalies or extending the model to accommodate additional sensor inputs.
5.Referecne issues:
Several relative research works are suggested to mention: https://doi.org/10.1016/j.eswa.2025.127067
Recommendation:
The paper presents a well-structured and innovative approach to flight trajectory prediction. However, minor revisions are necessary to enhance clarity, provide additional baseline comparisons, and refine result interpretations. The manuscript can be considered for publication after minor revision.
Author Response
Authors’ Responses to Reviewer 1’s Comments
Thank you for your careful reading our original submission. We truly appreciate your valuable feedback and constructive comments. In this current version, we have made effort to address all of your comments.
Gratefully, we believe that the paper has been improved significantly thanks to your comments. In particular, we have made specific effort to address your comments as follows:
Referee remarks:
Major Comments:
Comments 1: Clarity of Methodology:
The explanation of the IMM estimation process in Section 3.2 is relatively technical and may be difficult for readers unfamiliar with multiple model estimation techniques. It is recommended to provide a brief intuitive explanation before diving into the mathematical formulation.
The fusion strategy between IMM and Informer in Section 3.4 could be better structured. While the method is explained in detail, a flowchart illustrating the step-by-step prediction process would significantly enhance clarity.
Authors reply:
Thank you for your valuable suggestions. We have added an intuitive explanation before diving into the mathematical formulation, outlining its core ideas before the mathematical details. We have also added data flows to the Section 3.4 images to illustrate the IMM-Informer model, then the step-by-step prediction process explanations have been supplemented.
In the revised manuscript this change can be found –page 5, paragraph 4, line 194-201,
page 10, paragraph 1, line 318-334.
Comments 2: Comparative Analysis:
The study compares IMM-Informer with LSTM and standalone Informer models. However, it would be beneficial to include another baseline, such as a traditional Kalman filter-based approach, to further highlight the advantage of the hybrid methodology.
The justification for selecting LSTM and Informer as the primary deep learning baselines is not clearly stated. A brief discussion on why other deep learning models (e.g., Transformer-based models beyond Informer) were not considered would strengthen the experimental section.
Authors reply:
Thank you for your valuable suggestions. Kalman filtering requires the accurate state-space model which is difficult to obtain, and the motion state is often switched due to maneuvering during the flight phase. Therefore, the proposed flight trajectory prediction method mainly exploits IMM algorithm with multi-model filters interaction to obtain initial state prediction, and combines data-driven deep learning framework for error correction. Although it is difficult for a single model to accurately represent all motions in some flight phases, the combination of multiple typical models can often capture the flight dynamics switching. To verify the necessity of using a combination of multiple models for initial state prediction, the typical data-driven time series prediction deep learning models are used for comparison. LSTM is a specialized type of recurrent neural network that addresses gradient vanishing and explosion problems through the introduction of gating mechanisms, enabling it to capture long-term dependencies in sequences. Informer, like the Transformer, employs self-attention mechanisms and parallel computation to capture global dependencies in sequences. Unlike LSTM's autoregressive generation requiring iterative steps, Informer processes entire sequences in a whole. Informer is more efficient than traditional Transformer for the long sequence time-series forecasting tasks due to its ProbSparse self-attention mechanism and self-attention distilling mechanisms. Since LSTM and Informer are representative, they are chosen for comparison with the proposed method. Relevant explanations have been added in Section 4.4, and more comparative analysis of different methods has been added in the discussion section. These will better demonstrate the superiority of proposed IMM-Informer framework and shed light on the improvements brought about by its hybrid structure.
In the revised manuscript this change can be found –page 14, paragraph 4, line 448-457.
Comments 3: Evaluation Metrics and Result Interpretation:
While the experimental results in Table 2 and Table 3 show that the proposed approach outperforms the baselines, the magnitude of improvement in some phases is relatively small. A discussion on the practical significance of these improvements for air traffic management applications is suggested.
The MAE, RMSE, and MAPE metrics are appropriate for evaluating trajectory prediction accuracy, but additional statistical significance testing (e.g., paired t-tests or confidence intervals) could further confirm the robustness of the performance gains.
Authors reply:
Thank you for pointing this out. In fact, the units in Table 2 and Table 3 are different, so some of the margins of error appear to be smaller, and the units are unified in Table 3. In the revision, we have added units to the table and added descriptions. Table 2 and Table 3 show that the proposed approach outperforms the baselines, the smaller the errors are with the real trajectory, the higher the accuracy of trajectory prediction. In actual ATC system, accurate flight path prediction can well present the current and future air operation situation, and provide decision making basis for the identification potential air conflicts and optimization path by controllers. The relevant explanations have been added. In addition, we also added confidence intervals to show the significance of the predicted results.
In the revised manuscript this change can be found –page 16, Table 2, page 17, Table 3, and page 19, Figure 6; page 16, paragraph 1, line 488-490.
Minor Comments:
Comments 4: Abstract Length:
The abstract is informative but slightly lengthy. Consider streamlining the explanation of the methodology while retaining key results to improve conciseness.
Authors reply:
We appreciate your feedback and agree that the abstract could be streamlined to make it more concise. In the revised version, we have simplified the description while retaining the core workflow logic and highlighting the key results to emphasize impact. This improvement will enhance readability and bring the abstracts in line with journal standards.
In the revised manuscript this change can be found –page 1, Abstract.
Comments 5: Equation Formatting Issues:
Please check whether there are formatting inconsistencies around Equation (21) in Section 3.3. The indentation or spacing might need adjustments for better readability.
Authors reply:
Thank you for your careful examination. The formatting inconsistencies in Section 3.3 have been revised in the manuscript.
In the revised manuscript this change can be found –page 7, paragraph 4, line 245; page 8, paragraph 2, 3 and 4, line 265, 271 and 286.
Comments 6: Figures and Tables:
The trajectory prediction results in Figure 4 effectively visualize the model's performance. However, adding a zoomed-in comparison of key regions where the predictions diverge from the actual trajectory could provide more insight.
Authors reply:
Thank you for pointing this out. The relevant parts of the Figure 4 have been revised, and the zoomed-in comparison of key regions where the predictions diverge from the actual trajectory have been added in Figure 6.
In the revised manuscript this change can be found –page 15 and 19, Figure 4 and Figure 6.
Comments 7: Conclusion Section:
The conclusion effectively summarizes the study but could briefly mention potential real-world deployment challenges and future research directions, such as handling ADS-B data anomalies or extending the model to accommodate additional sensor inputs.
Authors reply:
Thank you for pointing this out. Potential challenges in practical applications and future research directions have been added to the conclusions.
In the revised manuscript this change can be found–page 20, the Section of Conclusions, paragraph 2.
Comments 8: Reference issues: Several relative research works are suggested to mention:
https://doi.org/10.1016/j.eswa.2025.127067
Authors reply:
Thank you for your suggestion. Relevant content has been added in the Section of Related work.
In the revised manuscript this change can be found –page 4, paragraph 2, line 146-149, and [32].
Finally, we highly appreciate the opportunity to revise and resubmit the manuscript. All your comments and suggestions have helped us improve the manuscript significantly. We hope we have answered all your questions and have revised the manuscript to your satisfaction. with this minor revision, we hope you could give the study a fresh assessment.

Reviewer 2 Report
Comments and Suggestions for Authors
The paper presents a hybrid framework for flight trajectory prediction using a combination of Interacting Multiple Models (IMM) and an Informer-based deep learning model. While the idea of fusing model-based state estimation with data-driven correction is interesting, the paper suffers from several major weaknesses that undermine the strength of its contributions. The following issues are identified:
The combination of model-based state estimation (IMM) with data-driven deep learning (Informer) is not fundamentally new. Similar hybrid approaches have been explored in trajectory prediction for both air traffic and autonomous driving. For example, Kalman filtering and particle filtering have been integrated with LSTM-based models in several prior works, yet the paper does not explain why using Informer instead of other sequence models leads to better performance. Without a clear theoretical or methodological advancement, the contribution is reduced to an application of existing techniques to a new problem domain rather than a novel technical innovation.
Also IMM-KF has been published on IROS 2024 (which also use CV CA CT model), author failed cite this paper "Nguyen, An Duy, et al. "A Multi-model Fusion of LiDAR-inertial Odometry via Localization and Mapping." 2024 IEEE/RSJ International Conference on Intelligent Robots and Systems (IROS). IEEE, 2024."
The figures in the paper are poorly designed and lack necessary details. Axis labels are missing or unclear in several plots, and no units are provided for the prediction error values. The color legend is not well defined, and there is no explanation of why certain patterns or deviations are observed in the trajectory plots. Furthermore, Figure 3 (which presents the hybrid architecture) does not sufficiently illustrate the data flow between the IMM and the Informer model, making it difficult to understand the fusion process. A well-designed figure should help clarify the methodology rather than confuse the reader.
The paper fails to adequately reference prior work [1-20] in trajectory prediction and state estimation. While a few trajectory prediction papers are cited, key works involving hybrid filtering and learning-based models are missing. For example, the authors fail to mention classical Kalman filtering approaches, Social GAN for human trajectory prediction, and hybrid state estimation models used in autonomous driving. This weakens the paper's positioning in the existing research landscape and makes the contribution appear less significant. A more thorough literature review would strengthen the case for why this specific IMM-Informer combination is necessary.
The paper uses a proprietary dataset of ADS-B signals, but no experimentation is performed on publicly available datasets for trajectory prediction (e.g., OpenSky Network or NASA flight trajectory data) or local area simulated dataset with Bacon can also be considered (e.g., MCD dataset and NTU-Viral Dataset) . This makes it difficult to evaluate the generalization ability of the model and to compare it with other state-of-the-art methods. Testing on a well-known benchmark dataset would allow for direct performance comparison with existing work and provide more credibility to the reported improvements.
The choice of baseline models is insufficient to demonstrate the superiority of the proposed method. The paper compares the hybrid IMM-Informer model only with a standalone Informer and an LSTM model. However, standard Kalman filters, Extended Kalman Filters (EKF), Unscented Kalman Filters (UKF), and even particle filters should be included as baselines, since these are commonly used for trajectory prediction in dynamic environments. The lack of comprehensive comparison makes it unclear whether the reported improvements are due to the hybrid nature of the model or simply the use of Informer instead of a conventional RNN model.
The choice of Informer over other Transformer-based architectures (e.g., Linformer, Reformer, Longformer) is not well justified. Informer is known for handling very long sequences, but flight trajectory prediction typically involves short- to medium-term predictions. It is unclear why Informer would perform better than a standard Transformer, LSTM, or GRU in this context. A head-to-head comparison between Informer and other Transformer variants would strengthen the claim that Informer is particularly well-suited for this task.
The paper presents error metrics (e.g., RMSE, MAE) but lacks deeper analysis of model failure cases and robustness under difficult conditions. For example, how does the model perform during rapid altitude changes, sensor signal loss, or GPS jamming? A detailed error breakdown across different flight phases (e.g., ascent, descent, cruise) would provide insight into the model’s strengths and weaknesses. Additionally, the absence of confidence intervals or statistical significance tests raises concerns about the reproducibility of the reported improvements.
While the model is trained and tested on real-world ADS-B data, the authors do not demonstrate how well the model generalizes to different routes, weather conditions, or aircraft types. The paper does not report performance across multiple airports or airlines, which would be necessary to confirm that the model is not overfitting to a specific dataset. Furthermore, the computational cost and real-time feasibility of deploying the model in an actual air traffic control system are not discussed.
The training procedure and hyperparameter tuning strategy are poorly documented. The paper does not state how the model’s hyperparameters (e.g., learning rate, batch size) were chosen or how many epochs were used during training. Furthermore, it is unclear whether the training procedure included any form of early stopping or regularization to prevent overfitting. Without this information, the reported improvements could be due to overfitting rather than genuine model superiority.
The paper mentions that ADS-B data from multiple sources are fused for trajectory prediction, but the details of this process are vague. How are the signals aligned in time and space? How does the model handle inconsistent or missing signals? Are any filtering or smoothing techniques applied before feeding the data into the model? A detailed explanation of the data preprocessing and fusion strategy is necessary to ensure the results are reproducible and robust.
The paper suffers from grammatical errors, poor sentence structure, and inconsistent terminology. Technical terms are introduced without clear definitions, and the mathematical notation is not consistent throughout the paper. The writing style is overly dense in some sections and too superficial in others. Additionally, the transition between sections is abrupt, making the paper difficult to follow. A thorough language edit and a more structured flow would significantly improve the readability and professionalism of the paper.
The paper mentions that improved flight trajectory prediction can enhance air traffic control, but the broader implications are not explored. How does the model improve airspace capacity or reduce flight delays? Does it help in fuel efficiency or emission reduction? What are the potential regulatory and safety challenges in deploying this model? A discussion of the broader impact and limitations would make the paper more comprehensive and practically relevant.
The paper needs major revision, Particularly on reference. The idea of IMM has been used before, so the author should cite it. For other similar works,the author should also cite them and if possible conduct comparisons. Some of them already open-sourced and also use becon based approach.
https://github.com/brytsknguyen/uloc
https://github.com/KIT-ISAS/SFUISE
https://github.com/sair-lab/localization
[1] Li, Jiarong, et al. "Vlocsense: Integrated vlc system
for indoor passive localization and human sensing."
Proceedings of the 30th Annual International Conference on
Mobile Computing and Networking. 2024.
[2] Zhao, Chenyu, et al. "Foes or Friends: Embracing Ground
Effect for Edge Detection on Lightweight Drones."
Proceedings of the 30th Annual International Conference on
Mobile Computing and Networking. 2024.
[3] Wang, Chen, et al. "Ultra-wideband aided fast
localization and mapping system." 2017 IEEE/RSJ
international conference on intelligent robots and systems
(IROS). IEEE, 2017.
[4] Lim, Hyungtae, Changgue Park, and Hyun Myung. "RONet:
Real-time range-only indoor localization via stacked
bidirectional LSTM with residual attention." 2019 IEEE/RSJ
International Conference on Intelligent Robots and Systems
(IROS). IEEE, 2019.
[5] Yuan, Shenghai, et al. "Large-Scale UWB Anchor
Calibration and One-Shot Localization Using Gaussian
Process." arXiv preprint arXiv:2412.16880 (2024).
[6] Song, Yang, et al. "Uwb/lidar fusion for cooperative
range-only slam." 2019 international conference on robotics
and automation (ICRA). IEEE, 2019.
[7] Zhou, Haoyu, Zheng Yao, and Mingquan Lu. "Lidar/UWB
fusion based SLAM with anti-degeneration capability." IEEE
Transactions on Vehicular Technology 70.1 (2020): 820-830.
[8] Chen, Zhijian, et al. "NLOS identification-and
correction-focused fusion of UWB and LiDAR-SLAM based on
factor graph optimization for high-precision positioning
with reduced drift." Remote Sensing 14.17 (2022): 4258.
[9] Sun, Jian, et al. "A novel UWB/IMU/odometer-based
robot localization system in LOS/NLOS mixed environments."
IEEE Transactions on Instrumentation and Measurement
(2024).
[10] Zhou, Boli, Hongbin Fang, and Jian Xu.
"UWB-IMU-odometer fusion localization scheme: Observability
analysis and experiments." IEEE Sensors Journal 23.3
(2022): 2550-2564.
[11] Sun, Jian, et al. "UWB-IMU-Odometer Fusion for
Simultaneous Calibration and Localization." IEEE Internet
of Things Journal (2024).
[12] Chen, Liangming, et al. "Relative localizability and
localization for multi-robot systems." IEEE Transactions on
Robotics (2025).
[13] Barral, Valentín, Carlos J. Escudero, and José A.
García-Naya. "NLOS classification based on RSS and ranging
statistics obtained from low-cost UWB devices." 2019 27th
European Signal Processing Conference (EUSIPCO). IEEE,
2019.
[14] Ozeki, Tomohiro, and Nobuaki Kubo. "Gnss nlos signal
classification based on machine learning and pseudorange
residual check." Frontiers in Robotics and AI 9 (2022):
868608.
[15] Yang, Hongchao, et al. "UWB sensor-based indoor
LOS/NLOS localization with support vector machine
learning." IEEE Sensors Journal 23.3 (2023): 2988-3004.
[16] Li, Kailai, Ziyu Cao, and Uwe D. Hanebeck.
"Continuous-time ultra-wideband-inertial fusion." IEEE
Robotics and Automation Letters 8.7 (2023): 4338-4345.
https://github.com/KIT-ISAS/SFUISE
Comments on the Quality of English Language
Nil
Author Response
Authors’ Responses to Reviewer 2’s Comments
Thank you for your careful reading our original submission. We truly appreciate your valuable feedback and constructive comments. Hope you may grant us another consideration for this revision.
In this current version, we have made every effort to address all of your comments. Gratefully, we believe that the paper has been improved significantly thanks to your comments. We hope you would grant this improved version a second thought with a fresh assessment. In particular, we have made specific effort to address your comments as follows:
Referee remarks:
Comments 1: The combination of model-based state estimation (IMM) with data-driven deep learning (Informer) is not fundamentally new. Similar hybrid approaches have been explored in trajectory prediction for both air traffic and autonomous driving. For example, Kalman filtering and particle filtering have been integrated with LSTM-based models in several prior works, yet the paper does not explain why using Informer instead of other sequence models leads to better performance. Without a clear theoretical or methodological advancement, the contribution is reduced to an application of existing techniques to a new problem domain rather than a novel technical innovation.
Authors reply: Thank you for your suggestions. We agree with this comment that there are approaches to combine Kalman filtering or particle filtering with time series forecasting network like LSTM. But the motivation proposed and the problem addressed are different. Kalman filtering or particle filtering requires an accurate state-space model, while LSTM is designed to capture long-term dependencies in time series, but when processing the long track data in seconds in this paper, not only the long-term prediction performance is inadequate, but also the problem of gradient explosion may occur in LSTM. Informer introduced the ProbSparse self-attention mechanism, which significantly reduced the computational complexity of the self-attention mechanism, which enabled it to efficiently process longer sequences, better capture the remote dependencies in the flight surveillance data, and thus improve the prediction accuracy. Compared to existing studies, our work stands out in several aspects: (1) we adopted an innovative hybrid method by integrating multiple dynamic models with an effective generative model for long-term sequence prediction, which can further improve the accuracy of trajectory prediction; (2) we addressed an under explored angle of fusion mode between dynamics modeling and learning-based method, where the initialization and prediction model training using IMM and Informer, thus the flight trajectory prediction is realized by the state prediction compensation with the correction error sequence; (3) experiments based on real received flight surveillance sensor data verify the effectiveness and feasibility of the proposed method. Relevant contents have been supplemented in the revised manuscript.
In the revised manuscript this change can be found –page 18, paragraph 3, line 528-539,
page 19, paragraph 2, line 540-553,
page 20, paragraph 3, line 581-586.
Comments 2: Also IMM-KF has been published on IROS 2024 (which also use CV CA CT model), author failed cite this paper "Nguyen, An Duy, et al. "A Multi-model Fusion of LiDAR-inertial Odometry via Localization and Mapping." 2024 IEEE/RSJ International Conference on Intelligent Robots and Systems (IROS). IEEE, 2024."
Authors reply: Thank you for pointing this out. Relevant content has been added in the section of Related work.
In the revised manuscript this change can be found –page 4, paragraph 1, line 114-117, "For example, Nguyen et al. [17] proposed multi-model fusion LiDAR-inertial odometry method incorporated the Interactive Multiple Models and Kalman Filter (IMMKF), which can demonstrate superior accuracy for reliable navigation in dynamic motion and noisy conditions."
Comments 3: The figures in the paper are poorly designed and lack necessary details. Axis labels are missing or unclear in several plots, and no units are provided for the prediction error values. The color legend is not well defined, and there is no explanation of why certain patterns or deviations are observed in the trajectory plots. Furthermore, Figure 3 (which presents the hybrid architecture) does not sufficiently illustrate the data flow between the IMM and the Informer model, making it difficult to understand the fusion process. A well-designed figure should help clarify the methodology rather than confuse the reader.
Authors reply: Thank you for pointing this out. In the revision, some figures in this paper have been updated and added details, and the data units have been checked. The prediction results for the three dimensions of latitude, longitude, and altitude are shown in Figure 4, but the data are consistent in time, so the same time coordinates are used. In Figure 3 we have added the data flow between the inputs and outputs of modules in a clear and detailed way. Since the proposed method makes full use of the multiple dynamics models to capture the kinematic switching, then uses this as the initialization of the data-driven method, combined with historical data to train the trajectory prediction model, the prediction performance will be improved in some key phases compared with the deep learning framework alone. Relevant descriptions have been supplemented in the comparation analysis of the results.
In the revised manuscript this change can be found –page 10, Figure 3 and line 318-334,
page 15, paragraph 2, line 470-474.
Comments 4: The paper fails to adequately reference prior work [1-20] in trajectory prediction and state estimation. While a few trajectory prediction papers are cited, key works involving hybrid filtering and learning-based models are missing. For example, the authors fail to mention classical Kalman filtering approaches, Social GAN for human trajectory prediction, and hybrid state estimation models used in autonomous driving. This weakens the paper's positioning in the existing research landscape and makes the contribution appear less significant. A more thorough literature review would strengthen the case for why this specific IMM-Informer combination is necessary.
Authors reply: Thank you for pointing this out. In this paper, we review the current research on flight trajectory prediction. In the revision, we first added the description of typical Kalman filtering, IMM filtering and their earlier related studies in air traffic control tracking, emphasizing the dependence of traditional state-space model-based methods on accurate models. We also supplement the development of hybrid filter in fields such as human trajectory sensing, autonomous driving and mobile robots, as well as some recent representative studies such as IMMKF. In addition, we also cite the model-driven and data-driven integration methods, and introduce some recent representative studies, such as physics-informed LSTM-based aircraft dynamic force identification. The researchers also found that uncertainty often exists in the models for complex sensor scenarios. At this time, the hybrid method reflects the advantages compared with the single network learning-based method. These supplements reinforce the necessity of the specific hybrid method proposed in this paper for the uncertain model.
In the revised manuscript this change can be found –page 3, paragraph 3, line 100-105,
page 3, paragraph 3, line 111-117,
page 4, paragraph 1, line 144-149,
page 4, paragraph 2, line 152-155.
Comments 5: The paper uses a proprietary dataset of ADS-B signals, but no experimentation is performed on publicly available datasets for trajectory prediction (e.g., OpenSky Network or NASA flight trajectory data) or local area simulated dataset with Bacon can also be considered (e.g., MCD dataset and NTU-Viral Dataset). This makes it difficult to evaluate the generalization ability of the model and to compare it with other state-of-the-art methods. Testing on a well-known benchmark dataset would allow for direct performance comparison with existing work and provide more credibility to the reported improvements.
Authors reply: Thank you for your suggestions. The proposed method is a kind of hybrid method which combines dynamic modeling and data-driven method. Different from existing fully data-driven methods, this method needs to make use of certain prior knowledge of dynamic processes models. Therefore, it needs to utilize the flight route information known to ADS-B flight surveillance data, which is also in line with the actual sensor-based flight surveillance in the terminal area. However, some public datasets, such as NASA flight trajectory data, cannot effectively obtain the prior dynamics information corresponding to the data, which do not meet some assumptions of the proposed method. The main motivation of the proposed method is how to effectively use certain imprecise dynamic model prior information combined with sensor flight surveillance data for flight trajectory prediction. For the generalization performance of the algorithm, we have also added relevant comparison experiments, and the results have been discussed and analyzed with the data of different routes used for verification. In addition, the premise assumptions of the proposed method and the challenges for the general dynamics modeling have been supplemented in the revised manuscript.
In the revised manuscript this change can be found –page 18, paragraph 2, line 510-526,
page 19, paragraph 2, line 554-559.
Comments 6: The choice of baseline models is insufficient to demonstrate the superiority of the proposed method. The paper compares the hybrid IMM-Informer model only with a standalone Informer and an LSTM model. However, standard Kalman filters, Extended Kalman Filters (EKF), Unscented Kalman Filters (UKF), and even particle filters should be included as baselines, since these are commonly used for trajectory prediction in dynamic environments. The lack of comprehensive comparison makes it unclear whether the reported improvements are due to the hybrid nature of the model or simply the use of Informer instead of a conventional RNN model.
Authors reply: Thank you for your suggestions. Although Kalman filter and its extension such as EKF, UKF and particle filters are often used for state prediction with dynamic system, this kind of based state estimation method needs to assume that the real motion model is known, that is, the state space model can be accurately obtained. However, the accurate state space model is often difficult to obtain in practice, and only certain inaccurate dynamic models can be exploited, and flight trajectory prediction can be conducted by using flight surveillance data. The original experiment only illustrates the performance improvement of the proposed method over the standalone data-driven method under the same assumptions. To further illustrate the effectiveness of the proposed method, we have added the baselines, including comparisons of different hybrid approaches with the same assumptions as the proposed method, including the recurrent network models such as GRU and LSTM used for learning, as well as comparisons with classical Kalman filter that assume known models, to comprehensively analyze the performance of the proposed method. Relevant content has been added in the Section of Discussion.
In the revised manuscript this change can be found –page 18, paragraph 2, line 519-526,
page 19, paragraph 2 , line 540-549.
Comments 7: The choice of Informer over other Transformer-based architectures (e.g., Linformer, Reformer, Longformer) is not well justified. Informer is known for handling very long sequences, but flight trajectory prediction typically involves short- to medium-term predictions. It is unclear why Informer would perform better than a standard Transformer, LSTM, or GRU in this context. A head-to-head comparison between Informer and other Transformer variants would strengthen the claim that Informer is particularly well-suited for this task.
Authors reply: Thank you for your suggestions. In the revision, we added the hybrid methods that combines different sequence forecast network models for comparison. Among them, IMM-LSTM and IMM-GRU represent the fusion of data-driven methods based on traditional RNN architecture with IMM estimation. IMM-Transformer represents the approach based on the Transformer architecture. The reason for choosing Informer is that it is a general optimal solution for long time series prediction with ProbSparse self-attention mechanism, especially in the balance between efficiency and precision. While Transformer itself represents a scenario where global attention is appropriate, other Transformer variants offer advantages only for specific hardware or data tasks. The relevant extended comparisons and analysis have been supplemented in the Section of Discussion.
In the revised manuscript this change can be found –page 18, paragraph 2, line 519-526,
page 19, paragraph 2, line 549-553.
Comments 8: The paper presents error metrics (e.g., RMSE, MAE) but lacks deeper analysis of model failure cases and robustness under difficult conditions. For example, how does the model perform during rapid altitude changes, sensor signal loss, or GPS jamming? A detailed error breakdown across different flight phases (e.g., ascent, descent, cruise) would provide insight into the model’s strengths and weaknesses. Additionally, the absence of confidence intervals or statistical significance tests raises concerns about the reproducibility of the reported improvements.
Authors reply:
Thank you for your suggestions. The dataset used in the experiment is based on the flight surveillance data received by the sensors, which contains the information such as weather, rapid state changes that may be brought about by complex flight conditions, sensor signal loss, anomalies and noise. The packet dropout, abnormal points and so on have been eliminated by data preprocessing. The proposed hybrid method is developed to exploit multiple basic models along with underlying Markov chain to capture the current possible dynamic changes. The use of multiple basic models rather than using some more accurate modeling such as considering random packet dropout or under rapidly changing conditions makes the proposed method robust, but interpretability becomes a challenge. In the revision, we have mentioned in the conclusions that designing effective generic models or model selection for more complex conditions is another important challenge that requires more prior information. We also added the detailed error breakdown for different flight phases, and the confidence intervals with trajectory prediction have been included to show the significance and reproducibility of the results.
In the revised manuscript this change can be found –page 19, Figure 6,
page 19, paragraph 2, line 540-543, line 554-559,
page 20, Figure 7; paragraph 2, line 564-572.
Comments 9: While the model is trained and tested on real-world ADS-B data, the authors do not demonstrate how well the model generalizes to different routes, weather conditions, or aircraft types. The paper does not report performance across multiple airports or airlines, which would be necessary to confirm that the model is not overfitting to a specific dataset. Furthermore, the computational cost and real-time feasibility of deploying the model in an actual air traffic control system are not discussed.
Authors reply: Thank you for pointing this out. Although the ADS-B flight surveillance data used in the experiment is the certain route, the data contains a variety of real flight trajectories under different aircraft types and different weather conditions. We agree the importance of generalization or overfitting issues for algorithm. In the revision, we have supplemented the discussion of relevant added experiments. The training dataset has been expanded by the flight surveillance data of multiple flights with different routes and the flight trajectories generated by the stochastic system model, which was exploited to analyze and discuss the generalization of the proposed method with multiple baselines comparison. In addition, we also added the efficiency comparison of the algorithms, and discuss the real-time and feasibility of deploying the proposed method in real air traffic control systems. The relevant content has been added to the discussion section.
In the revised manuscript this change can be found –page 18, paragraph 2, line 510-518,
page 20, paragraph 2, line 568-572.
Comments 10: The training procedure and hyperparameter tuning strategy are poorly documented. The paper does not state how the model’s hyperparameters (e.g., learning rate, batch size) were chosen or how many epochs were used during training. Furthermore, it is unclear whether the training procedure included any form of early stopping or regularization to prevent overfitting. Without this information, the reported improvements could be due to overfitting rather than genuine model superiority.
Authors reply:
We agree with this comment. We understand the importance of hyperparameters tuning. In the revised paper, we have added the detailed description of the training and validation procedure, including how hyperparameters like learning rate and batch size were selected, the number of epochs used, and the application of techniques such as early stopping to prevent overfitting. This will enhance the transparency and credibility of our results.
In the revised manuscript this change can be found –page 13, paragraph 3, line 420-422,
page 13, paragraph 3, line 425-427,
page 14, Table 1.
Comments 11: The paper mentions that ADS-B data from multiple sources are fused for trajectory prediction, but the details of this process are vague. How are the signals aligned in time and space? How does the model handle inconsistent or missing signals? Are any filtering or smoothing techniques applied before feeding the data into the model? A detailed explanation of the data preprocessing and fusion strategy is necessary to ensure the results are reproducible and robust.
Authors reply:
Thank you for your suggestions. The ADS-B data fusion mentioned in this paper mainly refers to the algorithm-based fusion, which integrates multiple dynamic model information into historical sensor data for model training and verification. The flight surveillance data used here is the time series data from the ADS-B messages received by the sensor after decoding. For abnormal or missing signals, techniques such as interpolation are used to pre-process the data before it is fed into the model, including the following
Time-Space Alignment: ADS-B data from multiple sensors are synchronized via timestamp matching (±1s tolerance). Spatial alignment uses Mercator projection to convert geodetic coordinates (latitude/longitude) into a unified Cartesian frame, ensuring consistent spatial referencing across sensors.
Handling Inconsistencies or missing data: prior to modeling, we apply a two-step smoothing process. The first is outlier removal, in which the data points exceeding 3σ from the local mean are flagged and removed. The second is interpolation, in which the missing values are interpolated using cubic spline interpolation.
Normalization: features are scaled to [0,1] using Min-Max normalization.
These supplemental instructions can be found in the revised manuscript, which ensure robustness with data and improve reproducibility.
In the revised manuscript this change can be found –page 11, paragraph 2, line 359- 367.
Comments 12: The paper suffers from grammatical errors, poor sentence structure, and inconsistent terminology. Technical terms are introduced without clear definitions, and the mathematical notation is not consistent throughout the paper. The writing style is overly dense in some sections and too superficial in others. Additionally, the transition between sections is abrupt, making the paper difficult to follow. A thorough language edit and a more structured flow would significantly improve the readability and professionalism of the paper.
Authors reply:
Thank you for pointing this out. We have carefully revised the paper for various grammatical errors, poor sentence structure and terminological inconsistencies. In addition, the paper writing has been improved to keep the writing style consistent throughout the article.
In the revised manuscript this change can be found –page 1, Section of Abstract,
page 4, paragraph 3, line 152-155,
page 5, paragraph 4, line 194-201,
page 11, paragraph 2, line 359-367,
page 14, paragraph 4, line 448-458,
page 20, paragraph 3, line 589-600.
Comments 13: The paper mentions that improved flight trajectory prediction can enhance air traffic control, but the broader implications are not explored. How does the model improve airspace capacity or reduce flight delays? Does it help in fuel efficiency or emission reduction? What are the potential regulatory and safety challenges in deploying this model? A discussion of the broader impact and limitations would make the paper more comprehensive and practically relevant.
Authors reply:
Thank you for your suggestions. In the revision, the impacts of improved flight trajectory prediction on ATC have been supplemented. In fact, by capturing the changing dynamics and temporal dependencies of flight states, the proposed method can provide air traffic control with more accurate information of trajectory prediction and more ample decision time in dense airspace to identify potential conflicts (e.g., cross routes, altitude layer occupancy) in advance, thus trajectory-based ATC decision making can be realized to reduce bypass flights with avoidance and indirectly increasing airspace capacity. In terms of fuel efficiency and emission reduction potential, the proposed method combined with dynamics representation and learning from historical sensors surveillance data generates predicted trajectories that can assist controllers in adjusting flight sequences in advance, reducing engine idle fuel consumption. The existing challenges and limitations include more general dynamics modeling in complex environments, which are also supplemented in the Section of Conclusion.
In the revised manuscript this change can be found –page 19, paragraph 2, line 554-559,
page 20, paragraph 3, line 589-600.
Comments 14: The paper needs major revision, Particularly on reference. The idea of IMM has been used before, so the author should cite it. For other similar works, the author should also cite them and if possible conduct comparisons. Some of them already open-sourced and also use becon based approach.
Authors reply:
Thank you for your suggestions. In the revision, Blom and Bar-Shalom’s pioneering work for the IMM algorithm has been cited in reference [12]. The related works for learning-based methods, hybrid methods have been updated, and the representative studies are also cited, which can be found in reference [15]-[17], [30]-[32]. In addition, the baselines have been extended, where the classical Kalman filtering and the hybrid methods such as IMM-LSTM,IMM-GRU and IMM-Transformer are compared with the proposed method. The advantages and challenges of the proposed method have been further discussed.
In the revised manuscript this change can be found –page 3, paragraph 3, line 100-105,
page 3, paragraph 3, line 111-117,
page 4, paragraph 1, line 144-149,
page 4, paragraph 2, line 152-155,
page 19, paragraph 2, line 548-553.

Round 2
Reviewer 2 Report
Comments and Suggestions for Authors
Author has addressed all of my concerns.
Comments on the Quality of English LanguageLanguage can be improved. There are many long lines.